# The Essential and Enigmatic Role of ABC Transporters in Bt Resistance of Noctuids and Other Insect Pests of Agriculture

**DOI:** 10.3390/insects12050389

**Published:** 2021-04-28

**Authors:** David G. Heckel

**Affiliations:** Max Planck Institute for Chemical Ecology, Hans-Knoell-Str. 8, D-07745 Jena, Germany; heckel@ice.mpg.de; Tel.: +49-3641-57-1599

**Keywords:** *Bacillus thuringiensis*, Cry protein, ATP-Binding Cassette, ABC Transporter, ATP switch model, pore-forming toxin, resistance, genetics, Noctuidae, *Helicoverpa*, *Spodoptera*, *Heliothis*, *Chloridea*, *Trichoplusia*

## Abstract

**Simple Summary:**

The insect family, Noctuidae, contains some of the most damaging pests of agriculture, including bollworms, budworms, and armyworms. Transgenic cotton and maize expressing Cry-type insecticidal proteins from *Bacillus thuringiensis* (Bt) are protected from such pests and greatly reduce the need for chemical insecticides. However, evolution of Bt resistance in the insects threatens the sustainability of this environmentally beneficial pest control strategy. Understanding the interaction between Bt toxins and their targets in the insect midgut is necessary to evaluate the risk of resistance evolution. ABC transporters, which in eukaryotes typically expel small molecules from cells, have recently been proposed as a target for the pore-forming Cry toxins. Here we review the literature surrounding this hypothesis in noctuids and other insects. Appreciation of the critical role of ABC transporters will be useful in discovering counterstrategies to resistance, which is already evolving in some field populations of noctuids and other insects.

**Abstract:**

In the last ten years, ABC transporters have emerged as unexpected yet significant contributors to pest resistance to insecticidal pore-forming proteins from *Bacillus thuringiensis* (Bt). Evidence includes the presence of mutations in resistant insects, heterologous expression to probe interactions with the three-domain Cry toxins, and CRISPR/Cas9 knockouts. Yet the mechanisms by which ABC transporters facilitate pore formation remain obscure. The three major classes of Cry toxins used in agriculture have been found to target the three major classes of ABC transporters, which requires a mechanistic explanation. Many other families of bacterial pore-forming toxins exhibit conformational changes in their mode of action, which are not yet described for the Cry toxins. Three-dimensional structures of the relevant ABC transporters, the multimeric pore in the membrane, and other proteins that assist in the process are required to test the hypothesis that the ATP-switch mechanism provides a motive force that drives Cry toxins into the membrane. Knowledge of the mechanism of pore insertion will be required to combat the resistance that is now evolving in field populations of insects, including noctuids.

## 1. Introduction

ABC proteins are a huge and ancient superfamily of proteins that are defined by the presence of a domain called the ATP-binding cassette [1]. Binding to ATP causes this domain to change its shape, and this conformational change has been harnessed for many purposes. The ABCE1 protein is required for termination of protein translation, where its lever-like action separates the large and small ribosomal subunits [2]. The Rad50 protein is invoked when chromosomes are damaged, by clamping onto DNA like tweezers to bring the strands together for repair [3]. The ABC transporters possess membrane-spanning helices that change orientation to squeeze small molecules across membranes when the ABC domain binds to ATP [4]. The most recently discovered property of ABC proteins is their role in the mode of action of pore-forming three-domain Cry toxins from *Bacillus thuringiensis* (Bt), the subject of this review.

The Cry toxins are important in agriculture because cotton, maize, and other crops have been transformed with genes from Bt to enable the plants to manufacture enough toxins to kill certain pest caterpillars that feed on them. See [5] for a comprehensive list of Bt toxins. When ingested along with plant tissue, the toxins form pores in the larval midgut epithelium, lysing the cells and eventually killing the insect. In 2019, 108.7 million hectares of transgenic Bt crops were grown worldwide [6], greatly reducing the use of chemical insecticides. The sustainability of this strategy is threatened by the evolution of resistance, which is already appearing in field populations of some pest insects (see [7] for a useful overview). Studies of Bt-resistant insects provided the first evidence for ABC transporters in the Cry toxin mode of action.

Cry1-type toxins are particularly potent against polyphagous noctuid pests, primarily the heliothines and the armyworms. The cotton bollworm *Helicoverpa armigera* (Hübner) was formerly distributed in Asia, Africa, and Australia, but has recently invaded South America. Its resistance to chemical insecticides and its ability to hybridize with the native corn earworm *Helicoverpa zea* (Boddie) has promoted its spread. *H. zea* populations in North America are evolving resistance to Cry1Ab-expressing maize. *Chloridea* (formerly *Heliothis*) *virescens* (Fabricius), the tobacco budworm [8], is a pest of cotton and other crops that has been well-controlled by Bt-cotton up to now. We will use the genus name *Heliothis* in this review for continuity with the literature cited. The fall armyworm *Spodoptera frugiperda* (J. E. Smith) has recently invaded Africa, India, Asia, and Australia from its home in the New World to become a worldwide pest of maize. Its congeners, the beet armyworm *Spodoptera exigua* (Hübner) and cotton leafworm *Spodoptera litura* (Fabricius), are also polyphagous pests. The cabbage looper *Trichoplusia ni* (Hübner) has evolved resistance to Dipel in greenhouses. Cry1 toxins are also effective against non-noctuid Lepidoptera, such as the diamondback moth *Plutella xylostella* (L), a worldwide pest of crucifer crops and the first insect to evolve Bt resistance in open field populations, and the domesticated silkworm *Bombyx mori* (L), a model insect for which powerful genetic tools have been developed. Cry1 resistance has appeared in some populations of the European corn borer *Ostrinia nubilalis* (Hübner) and the Asian corn borer *Ostrinia furnacalis* (Guenée). Cry2 toxins are active against many of the same Lepidoptera. The pink bollworm *Pectinophora gossypiella* has evolved resistance to Cry2-expressing Bt-transgenic cotton. The Cry3 toxins are potent against Coleoptera, such as *Diabrotica virgifera* (LeConte), the corn rootworm. ABC transporters of these non-noctuids will also be discussed as they illustrate the common features of the mode of action of the three-domain Cry toxins. Cell lines from noctuids have also been important in Bt research, including Sl-HP from *S. litura*, Sf9 from *S. frugiperda*, Hi5 from *T. ni*, and QB-Ha-E5 from *H. armigera*.

Unfortunately, there is no standard nomenclature yet for insect ABC transporters, which is becoming a problem for comparative studies as more and more are implicated as Bt targets. Gene and protein names provided by automatic annotation at NCBI do not uniquely identify them or reveal their relationship to the ABC transporters described here. The most reliable point of reference is the genome of *B. mori*, the first to be sequenced among Lepidoptera. Figure 1 shows the genomic context of the *B. mori* orthologs of the proteins discussed in this review. Figure 2 shows their predicted membrane topology.

## 2. Initial Discoveries

ABC transporters were first discovered as targets of Bt toxins by positional cloning in strains of resistant insects. Despite decades of research on Bt resistance, there was no previous biochemical or physiological evidence that ABC transporters could be involved. The discovery of their role was a surprise, and only the independent efforts by three groups that converged on the same protein has convinced the research community that understanding their role in resistance is important, as described in several recent reviews [5,12,13,14,15,16,17,18,19,20,21,22,23]. Solving this problem holds the key to developing strategies to combat the increasing problem of Bt resistance.

A mutation in the ABCC2 protein was identified in a Cry1Ac-resistant strain of *H. virescens*, by positional cloning using markers from the early versions of the *B. mori* genome sequence, well before a genome sequence for *H. virescens* was available [24]. Evidence that this mutation was important for resistance came from mapping, binding studies, and an allele frequency change correlated with the increase of resistance over time [24]. Like most ABC mutations subsequently found in other species, and like the cadherin mutation previously found in *H. virescens* [25] it introduced a frameshift and prevented expression of the full-length protein in the membrane. This contrasts with many cases of chemical insecticide resistance, where deletion of the target would be lethal.

Even stronger evidence came from analysis of a more subtle mutation in the ABCC2 protein in *B. mori* [26]. Positional cloning using the genome sequence converged on ABCC2, but no incapacitating mutation could be found. Instead, the protein in the Cry1Ab-resistant strain had several amino acid substitutions and an insertion of a tyrosine in an extracellular loop. Germline transformation was used to prove that one copy of the susceptible allele inserted in the resistant genetic background could confer Cry1Ab-susceptibility on the resistant strain [26], and subsequent experiments [27] proved that the inserted tyrosine was necessary and sufficient for resistance.

A similar approach using bulked segregant analysis based on cDNA markers identified ABCC2 and its neighboring gene ABCC3 as contributing to Cry1Ac and Cry1Ca resistance in *S. exigua* [28]. In contrast to the previous mutations, ABCC2 carried a lesion in an intracellular nucleotide-binding domain (NBD). Suppression of ABCC2 and ABCC3 by RNA inhibition (RNAi) increased the tolerance of susceptible larvae to Cry1Ac and Cry1Ca. This independent and unbiased positional cloning approach extended the phenomenon to a third genus, and to a third type of mutation in ABCC2, proving that ABC transporters could no longer be ignored in the mode of action of Cry toxins.

## 3. Search for ABC Mutants in Resistant Strains from Field and Laboratory

These early findings motivated the search for the involvement of ABC transporters in other Bt-resistant strains of noctuids and other Lepidoptera. Comparative linkage mapping with markers, previously shown to be linked to ABCC2 in *H. virescens*, was used to identify a mutation that eliminated the last transmembrane domain of ABCC2 in Cry1Ac-resistant *P. xylostella* from Hawaii [29]. The same study localized Cry1Ac resistance in *T. ni* from British Columbia to a region containing ABCC2, although a specific mutation was not identified [29]; a different resistance mechanism, altered aminopeptidase expression, was also identified in the same strain of *T. ni* [30]. Mis-spliced transcripts of ABCC2 generating a truncated protein were found in a Cry1Ac-resistant strain of *H. armigera* from China [31]. Another comparative linkage mapping approach identified a genomic region containing ABCC2 in a laboratory-selected Cry1F-resistant strain of *O. nubilalis* from collections in the Corn Belt of the USA [32], although involvement of ABCC2 has not yet been confirmed in field-evolved Cry1F resistance in *O. nubilalis* from Nova Scotia [33]. In Puerto Rico, rapid appearance of Cry1F resistance in *S. frugiperda* stimulated withdrawal of the transgenic maize variety from the market, and was found to be associated with mutations in ABCC2 [34]. Additional mutations in *S. frugiperda* ABCC2 were associated with Cry1Fa and Cry1A.105 resistance in Puerto Rico [35] and Brazil [36]. Screening for some of these was included in surveys using DNA diagnostics for resistance to chemical insecticides as well as Bt [37,38].

The only member of the ABCB family to be investigated in Lepidoptera as a target of Cry toxins initially came to attention because it was down-regulated in a Cry1Ac-resistant strain of *P. xylostella*. PxABCB1 expression was found to be lower in other resistant strains, was further reduced by additional Cry1Ac selection, and was suppressed by RNAi in susceptible strains, which increased their tolerance to Cry1Ac [39].

## 4. Functional Studies

Heterologous expression of ABC transporters in insect cell lines has been extensively used to probe their function. The crucial role of the tyrosine insertion in loop 2 of *B. mori* ABCC2 in conferring Cry1Ab resistance was convincingly shown by expression in Sf9 cells [27]. The same study demonstrated the synergy of the cadherin BtR175 and ABCC2 for the first time; co-expression of both made the cells much more susceptible to Cry1Ab than either one alone. These results were recapitulated using proteins from *H. virescens* in Sf9 cells [40], with the added information that synergy was observed only when both proteins were expressed in the same cell; i.e., not from a mixture of cells expressing one or the other, as might be expected from the sequential binding hypothesis [41]. The mechanism of synergy was further probed by comparing the ability of the cadherin from *H. armigera* or *S. litura* to synergize Cry1Ac action on *H. armigera* ABCC2 expressed in Hi5 cells [42]. Although the *S. litura* cadherin was an ineffective synergist, when cadherin repeat 11 from *H. armigera* was swapped in, synergistic activity increased. The authors hypothesized that specific binding sites on the cadherin localized the toxin to a good position for interaction with ABCC2 in a species-specific manner [42]. A similar species-specific synergism with ABCC2 from *S. exigua* and the cadherin from *S. exigua* (but not *H. armigera*) was observed with the Cry1C toxin in Sf9 cells [43]. Domain-swapping between Cry1C and Cry1Ac was used to infer that domains II and III of Cry1Ac have different binding sites on ABCC2 of *S. exigua* [44]. ABCC2 from *S. frugiperda* expressed in Hi5 cells conferred sensitivity to Cry1Ab and Cry1Fa, while the cadherin did not, but synergism was not investigated in this study [45]. ABCC3 from *S. frugiperda* was also found to confer sensitivity to Cry1Ab and Cry1Fa under similar conditions [46].

A wide-ranging study explored the specificity of the toxin-target interaction by expressing ABC transporters from Lepidoptera, Coleoptera, Diptera, and humans in Sf9 cells and testing them with lepidopteran- or coleopteran-active toxins [47]. ABCC2 or ABCC3 from *B. mori* conferred sensitivity to Cry1Aa, but not Cry1Ca or Cry1Da. The latter two must have different, unknown targets because they are active on caterpillars of some lepidopteran species. Human and dipteran ABC transporters tested did not respond to lepidopteran- or coleopteran-active toxins.

*D. melanogaster* is not normally susceptible to Cry1Ac, but when ABCC2 was expressed in the midgut of transgenic larvae, Cry1Ac in the artificial diet killed them [48]. Moreover, the genome of *D. melanogaster* lacks the ortholog of the 12-cadherin domain protein found in all Lepidoptera, so the killing mechanism did not rely on the same type of synergism from the cadherin. However, synergism could be observed when the transgenic larvae were fed peptide fragments from lepidopteran cadherins along with Cry1Ac [48].

The most sensitive measurements of the interaction between Cry toxins and their receptors have been made using heterologous expression in *Xenopus* oocytes [49]. Messenger RNA experimentally injected into these huge cells is translated and the proteins (e.g., ABCC2 or cadherin) are incorporated into the egg membrane. This technique is often used to investigate the properties of ion channels using the voltage-clamp technique. The current through the channel is measured as a function of the experimentally-fixed voltage gradient across the membrane and the resulting graph characterizes the electrophysiological properties of the channel. In this case, the channel is the Cry toxin pore inserted into the membrane, which allows inward cation flux. The dynamics of current flow depend on the details of pore insertion and structure. Using this sensitive technique, it was shown that expression of the cadherin alone produced almost no current, expression of ABCC2 allowed abundant current, and expression of both produced even more current—the most convincing demonstration of synergism to date.

The mechanism of synergism is still obscure, but a number of hypotheses can be envisaged, which are not mutually exclusive (Figure 3). These can be classified into *trans*-acting mechanisms where synergism can occur when the cadherin and ABC transporter may be separated from each other, and *cis*-acting mechanisms where synergism requires a close physical interaction. According to the sequential binding hypothesis [41], toxin monomers bind to the cadherin and are further processed by cleavage of the N-terminal α1-helix, whereupon they form oligomeric pre-pores in solution (Figure 3A). However, toxin monomer binding to the cadherin is not an absolute requirement for toxicity; cadherin knockouts can still be killed by higher toxin amounts [24,50,51,52] and Cry1Ac is still lethal to *T. ni* dispite not being able to bind to the *T. ni* cadherin [53]. Synergism is due to presence of the cadherin, which speeds up a process that happens at a slower rate in its absence. Here the pre-pore structure can diffuse away from the cadherin to interact with a remote ABC transporter, so this mechanism is classified as *trans*-acting. If, however the cadherin traps the toxin and brings it to the ABC transporter, this would be a *cis*-acting mechanism (Figure 3B). This could be synergistic if there are many more cadherin molecules in the membrane than ABCC2 molecules. Another previously suggested *cis*-acting mechanism [40] is shown in Figure 3C, where the cadherin pulls the pre-pore away from the ABC transporter, enabling it to insert into the membrane and freeing up the ABC transporter for the next pre-pore.

## 5. Extracellular Loops

Most of the ABC protein is out of reach of Bt toxins approaching the cell, since the nucleotide binding domains are entirely cytoplasmic, and most of the transmembrane domains are buried within the lipid bilayer (Figure 2). Extracellular loops connecting adjacent transmembrane helices are very short in ABCC proteins, but larger in ABCB and especially ABCA proteins. Detailed analyses of the interaction between extracellular loops of *B. mori* ABC transporters and various domains of Cry1A toxins have been carried out by the group of Ryoichi Sato in Tokyo. Following the discovery that insertion of a single tyrosine in Loop 2 of ABCC2 conferred resistance to Cry1Ab in larvae, swapping other amino acids for the inserted tyrosine blocked pore formation in cells expressing the transporter, while amino acid substitutions at other positions in the non-inserted loop did not [54]. Thus, the size of the loop, rather than its amino acid composition, was the more important determinant of sensitivity. Domain-swapping within the toxin implicated Domain II as most important in this interaction. Subsequent mutagenesis of Domain II of Cry1Aa revealed a region that bound both to ABCC2 and another important receptor, the cadherin BtR175 [55]. While ABCC3 had much lower binding affinities to Cry1Aa and Cry1Ab than ABCC2, binding was increased in constructs containing partial loops from ABCC2 [56]. Another group pointed out an amino acid difference within Loop 1 of ABCC2 of *S. frugiperda* and *S. litura* that correlated with binding affinity to Cry1Ac, and they also replicated the binding difference by creating two versions of the *H. armigera* ABCC2, one with each amino acid [57].

## 6. Regulatory Changes

Regulatory changes involving ABC transporters were also found to confer resistance. In a Cry1Ac-resistant strain of *P. xylostella*, resistance mapped to the vicinity of ABCC2 but no disruptive mutations in the gene could be found [58]. Instead, the expression level of ABCC2 and ABCC3 was found to be controlled by the mitogen-activated protein kinase (MAPK) signaling pathway, with the MAP4K4 gene located close by on the same genomic scaffold, accounting for the mapping results. Constitutive expression of MAP4K4 in the resistant strain suppressed ABCC2, ABCC3, and alkaline phosphatase, another Cry1Ac-binding protein. RNA interference (RNAi) suppression of MAP4K4 transiently restored susceptibility by upregulating ABCC2 and ABCC3. Thus, resistance in this case was due to higher expression of a negative regulator of ABCC2 transcription.

The Forkhead Box Protein A (FOXA) transcription factor was found to stimulate transcription of ABCC2 and ABCC3 in *H. armigera*, as predicted from FOXA binding sites in the promoters [59]. RNAi silencing of FOXA downregulated ABCC2 and ABCC3 and increased the tolerance of susceptible larvae to Cry1Ac. Parallel results were obtained by expression in Sl-HP cells in the same study. A different study screened several members of the GATA transcription factor family from *H. armigera* and found that GATAe caused Sf9, QB-Ha-E5, and Hi5 cell lines to increase their expression of ABCC2, conferring greater Cry1Ac sensitivity [60]. If either mechanism were to be found in resistant strains from the field, resistance would be due to lower expression of a positive regulator of ABCC2 transcription.

Comparison of the ABCC2 coding sequence across many species of Lepidoptera identified a conserved region targeted by the microRNA miR-998-3p [61]. MicroRNAs bind to messenger RNAs in a sequence-specific fashion and target them for destruction or inhibit translation. Injection of an agomir (a chemically modified RNA that mimics the effect of the miRNA) into susceptible larvae of *P. xylostella*, *S. exigua*, or *H. armigera* increased their tolerance of Cry1Ac and decreased the abundance of ABCC2 mRNA. Injection of an antagomir (a single-stranded molecule designed to block the effect of the miRNA) into larvae of Cry1Ac-resistant *P. xylostella* reduced their tolerance of Cry1Ac and increased their ABCC2 mRNA levels.

## 7. Cell Lines

The influential colloid-osmotic lysis theory to explain how Cry toxins kill cells was developed using different cell lines that naturally differed in their susceptibilities to two different toxins [62]. It would be interesting to determine which ABC transporters are naturally expressed by those cells. Sl-HP cells from *S. litura* are susceptible to activated Cry1Ac even though *S. litura* larvae are not. ABCC3 was found to be expressed in this cell line, and RNA inhibition of ABCC3 decreased Sl-HP sensitivity to Cry1Ac [63]. In another study, comparison of Cry1Ac sensitivities of cell lines from different tissues produced the order midgut > fat body > ovary as expected, but unexpectedly fat body-derived cells were most susceptible to Cry2Ab toxin [64]. Surveys of heterologous expression of candidate targets in cell lines [65] should also take their native expression patterns into account.

## 8. Cry2A Toxins

An extensive sampling effort in Australia employing the F2 screen [66] yielded strains of *H. armigera* and *H. punctigera* with high levels of resistance to the Cry2Ab toxin. Linkage mapping with these strains revealed several different mutations in the ABCA2 gene that prevented expression of the full-length protein [67]. Unlike the ABCC proteins, ABCA2 has two very large ectodomains (Figure 2), and because the mutants are extremely resistant to Cry2Ab, it was speculated that the single ABCA2 protein functions similarly to the sequential binding model for the cadherin and ABCC2 [67]. Linkage mapping in a strain of *T. ni* that was selected with Dipel in British Columbia greenhouses [68] eventually resulted in the identification of a transposable element in ABCA2 conferring Cry2Ab resistance [69]. Mis-splicing mutants in ABCA2 were associated with Cry2Ab resistance in *P. gossypiella* in a laboratory-selected strain from Arizona and field populations from India [70]. Additional crosses confirmed these mutations but suggested that an additional, uncharacterized mechanism was also involved in Cry2Ab resistance in this species [71].

A different member of the ABCC family from *H. armigera* (GenBank Accession No. KY796050) was also found to bind Cry2Ab, identified by the authors as “HaABCC1” [72], although it is not the ortholog of the ABCC1 (BGIBMGA007737+38) on *B. mori* Chromosome 15 next to ABCC2 and ABCC3 described previously [24,26,28]. The ortholog of “HaABCC1” in *B. mori* is part of a small cluster on Chromosome 12 (Figure 1D) of ABCC proteins with 5 additional N-terminal transmembrane domains as well as two very large ectodomains, unlike the Chromosome 15 ABCC1-ABCC2-ABCC3 cluster in *B. mori* and other Lepidoptera (Figure 2). Although the authors speculated on the role of ABCC proteins in cross-resistance between Cry1Ac and Cry2Ab, no binding or toxicity studies were performed with Cry1Ac, and the strain of *H. armigera* used was susceptible to both Cry1Ac and Cry2Ab [72]. The use of a name already assigned to another gene has unfortunately created the false impression that the results are relevant to cross-resistance. Here we designate this gene as ABCC6 (Figure 1D and Figure 2).

## 9. CRISPR/Cas9 Knockouts

CRISPR/Cas9 knockouts provide a very useful tool to investigate gene function in non-model organisms. The first use of the technique to knock out ABC transporters in Lepidoptera targeted the half-transporter genes *white*, *scarlet*, and *ok* in *Helicoverpa armigera* [73]. These are homologs of the well-known pigment transporters *white*, *scarlet*, and *brown* in *Drosophila melanogaster*, and as expected, the knockouts affected adult eye color and larval skin pigmentation. Homozygotes for the *white* knockout, however, were embryonic lethal in *H. armigera* and in the milkweed bug [74], which was unexpected because these are viable in *Drosophila*, *Aedes*, and *Tribolium*. Lethality has also complicated the interpretation for some knockouts of Cry toxin targets.

Knockouts of ABCA2 in a susceptible strain of *H. armigera* conferred >100-fold resistance to Cry2Aa and Cry2Ab, and eliminated Cry2Ab binding to BBMV, but did not affect resistance or binding to Cry1Ac [75]. After mapping Cry2Ab resistance in *T. ni* to ABCA2, where a transib mobile element was found to disrupt the gene, either ABCA1 or ABCA2 were knocked out in a susceptible strain and only ABCA2 was found to affect Cry2Ab tolerance [69]. Knockouts of the ABCA2 gene in *B. mori* (using the TALEN technique) conferred Cry2A resistance on larvae, and heterologous expression of ABCA2 in HEK239 cells confirmed the absence of cross-resistance to Cry1A, Cry1Ca, Cry1Da, Cry1Fa, and Cry9Aa toxins [76].

In *P. xylostella*, a knockout of ABCC2 conferred 724-fold resistance to Cry1Ac and a knockout of ABCC3 conferred 413-fold resistance to the same toxin. Each knockout greatly reduced BBMV binding to Cry1Ac, but the double knockout was not made in this study [77]. Somewhat different results were obtained in another study of the same species [78], in which single knockouts were weakly resistant (~4-fold) and only the double knockout was >8000-fold resistant to Cry1Ac. So far there is no explanation for the differing results in *P. xylostella*. A study in *H. armigera*, however, produced results closer to the second study, in that single knockouts of ABCC2 and ABCC3 were weakly resistant to Cry1Ac, while the double knockout was >15,000-fold resistant [79].

Knocking out ABCC2 in *O. furnacalis* conferred >300-fold resistance to Cry1Fa but less than 10-fold resistance to Cry1Ab or Cry1Ac and no resistance to Cry1Aa [80]. Knocking out ABCC2 in *S. frugiperda* increased tolerance to either Cry1Fa or Cry1Ab >120-fold, while knocking out ABCC3 increased tolerance by a lesser amount (>16-fold); in this study the double knockout was reported to be lethal [46]. In *S. exigua*, an ambitious study created single knockouts of ABCC1, ABCC2, or ABCC3, as well as the cadherin and an aminopeptidase, and examined susceptibility to Cry1Ac, Cry1Fa, and Cry1Ca. Among the 15 pairwise comparisons, ABCC2 had a strong effect and the cadherin a weak effect on Cry1Ac or Cry1Fa tolerance, and ABCC2 also had a weak effect on Cry1Ca tolerance [52].

CRISPR/Cas9 knockout experiments are useful in confirming the role of a given ABC transporter in susceptibility to a given toxin, and when more than one knockout and more than one toxin are compared, in assessing their relative importance. The possibility of non-target effects needs more investigation, in order to reconcile studies where double knockouts are lethal with those where they confer even more resistance, since these studies make diametrically opposed recommendations for resistance management. Targeting the nucleotide binding domains would increase the probability of nontarget effects, since these are more highly conserved across ABC family members. In addition, no knock-ins have been reported yet for ABC transporters, as in the case of another Bt resistance gene, tetraspanin [81].

## 10. Negative Cross-Resistance with Chemical Insecticides

The intriguing possibility that mutations in ABC transporters could interfere with the insect’s ability to use them to rid itself of other toxins has motivated many recent studies. With the increasing use of Bt sprays and transgenic plants in the 1990s, the issue of cross-resistance between Bt and chemical insecticides had achieved some attention, but Bt resistance mechanisms had not been characterized at the molecular level. More recently, in 2016 when several chemical insecticides were screened against an ABCC2-mutant strain of *H. armigera*, abamectin and spineotram were more toxic compared with their activity against a Cry1Ac-susceptible strain [82]. Measurements of higher abamectin concentrations in mutant larvae and transfected cells were consistent with the bioassay results. RNAi silencing of ABCC2 decreased susceptibility to Cry1Ac and increased susceptibility to abamectin [82]. However, the selective differential exerted by abamectin on the Cry1Ac-resistant versus Cry1Ac-susceptible strains was small, and not all subsequent studies have confirmed the effect.

Single and double knockouts of ABCC2 and ABCC3 in *H. armigera* produced in a different study were not more susceptible to abamectin or spinetoram [79]. The knockout of ABCC2 in *O. furnacalis* was not more susceptible to abamectin or chlorantraniliprole [80]. On the other hand, single knockouts of ABCC2 or ABCC3 in *S. frugiperda* were more susceptible to abamectin and spinosad (while the double knockout was reported to be lethal and could not be compared) [46]. Another study on *S. frugiperda* found that a Cry1F-resistant strain isolated from the field with a frameshift mutation in ABCC2 had lower sensitivity to bifenthrin and higher sensitivity to spinetoram; yet when ABCC2 was knocked out in a different strain, Cry1F resistance increased 25-fold but sensitivity to chlorantraniliprole, bifenthrin, spinetoram, and acephate was unchanged [83]. Knocking out the P-glycoprotein ABCB1 in *S. exigua* increased, rather than decreased, susceptibility to abamectin and emamectin benzoate [84]; whether this protein is orthologous to the coleopteran ABCB1 (see below) has not been determined. Contradictory results were obtained in a study of ABCC2 in *P. xylostella* [85]; HEK-293 cells stably transformed with ABCC2 accumulated less avermectin, but down-regulating ABCC2 in vivo with RNAi had no effect on avermectin or chlorfenapyr tolerance.

Although results are suggestive in some studies, the changes in tolerance to the conventional insecticides examined are small and would not be useful in a resistance-breaking approach against ABCC mutations, as mortality of Bt-resistant insects would not be much greater than their Bt-susceptible counterparts. Only a few insecticides have been examined so far, and some with a greater effect are likely to be found eventually in a wider screen.

## 11. ABCB (P-glycoprotein) and Cry3 Toxins in Coleoptera

A different family of ABC transporters, the P-glycoproteins (ABCB), is involved in toxicity of the Cry3 toxins in Coleoptera. Linkage mapping in a strain of the poplar leaf beetle *Chrysomela tremula* (Fabricius) resistant to transgenic Cry3Aa-expressing poplar identified a frameshift in the ABCB1 protein, and heterologous expression of ABCB1 in Sf9 cells conferred susceptibility to Cry3Aa in vitro [86]. In the western corn rootworm *D. virgifera virgifera*, heterologous expression of the orthologous ABCB1 protein also conferred Cry3A sensitivity on Sf9 and HEK-293 cells in vitro, and an mCry3A-resistant strain was found to have deletions in the ABCB1 gene [87].

Whether the beetle ABCB1 genes are orthologous to PxABCB1 from *P. xylostella* mentioned in Section 3 above [39] is not known; Cry3 toxins were not experimentally tested in that study. The authors pointed out structural similarities among Cry1 and Cry3 toxins, and searched for but could not find PxABCB1 in the fragmented *Plutella* genome sequence. We have found that the *Bombyx* ortholog maps to Chromosome 15, about 1 MB away from the ABCC cluster (Figure 1B). More studies are required to confirm the involvement of the P-glycoproteins in Cry1A toxin interactions in Lepidoptera, and to establish the generality of the P-glycoproteins as targets of the beetle-active Cry3 toxins. Cross-resistance studies suggest the existence of additional, different targets in beetles [88].

## 12. Other ABC Transporters in Lepidoptera

Suppression of the *white* gene in *P. xylostella* by RNAi reduced Cry1Ac susceptibility but was not lethal [89]; as pointed out earlier, CRISPR/Cas9 knockouts of *white* were lethal in *H. armigera* [73]. Suppression of ABCH1 in *P. xylostella* caused larval mortality but did not affect Cry1Ac resistance [90]. A gene in *O. furnacalis* identified as ABCG was down-regulated in Cry1Ab- and Cry1Ac-resistant strains [91]; it is evidently not orthologous to either of the two genes in *P. xylostella*. No mutations in half-transporter ABCG or ABCH family genes have yet been identified in Bt-resistant Lepidoptera.

## 13. Hypotheses on the Mechanism of Pore Insertion

The lack of three-dimensional structures of the ABCC2 or ABCA2 proteins, the Cry toxin pore embedded in the membrane, and the toxin-binding region of the cadherin has inhibited the development of detailed hypotheses on the manner by which ABC transporters facilitate pore insertion. ABC transporters could simply be another binding site on the membrane surface, increasing the local toxin concentration and increasing the rate of pore insertion, due to a concentration effect. It has been hypothesized that ABCC2 facilitates the formation of the pre-pore oligomer, in a manner similar to the cadherin [92]. It was hypothesized that active opening and closing of the ABC transporter channel would be required to pull the pre-formed pore into the membrane [13]. This hypothesis would seem to be refuted by results with a mutant ABCC2 from *S. exigua* lacking the second nucleotide-binding domain [93], as well as engineered mutants of ABCC2 from *B. mori* lacking nucleotide-binding domains [94].

As previously pointed out [15], it is difficult to explain how evolution in *Bacillus thuringiensis* has resulted in Cry2A-type toxins that target ABCA proteins, Cry1A-type toxins that target ABCC proteins, and Cry3B-type toxins that target ABCB proteins, without invoking some fundamental property that unites these very different ABC transporters. If the shared ATP-switch mechanism powering substrate transport [4] is not such a property, then we are left without a mechanistic explanation of the pore-insertion process for the three-domain Cry proteins [13]. Many other bacterial pore-forming toxins enter the membrane with dynamic conformational changes, for example Vip3A [95] or the Membrane Attack Complex [96], or the Tc toxin [97]. Whether such a dynamic process is required for Cry toxin pore formation deserves more investigation. A reasonable hypothesis at this point is that the dynamism comes from the toxin-target interaction, not just the toxin.

## 14. Future Perspectives

Since the first description in 2010, mutations in ABC transporters have emerged as the most important type of mutation causing resistance to the three-domain Cry toxins of *Bacillus thuringiensis*. Yet mechanistic studies have lagged behind those on other pore-forming toxins with much more complicated structures. Why? Not enough effort has been expended on determining the three-dimensional structures that will be required for a full understanding of how the toxin interacts with membrane proteins to form a membrane pore. Since the first two structures of trypsin-activated monomers of the three-domain Cry toxins revealed their structural similarity [98,99], many more have appeared confirming that this similarity is fundamental. Recently a structure of the entire protoxin was determined [100], revealing additional domains that might potentiate pore formation in some way. The low-hanging fruit has been harvested. Lacking is a structure of any ABC transporter known to interact with a Cry toxin. Lacking is a structure of any cadherin known to interact with a Cry toxin. Lacking is a structure of the Cry toxin pore in the membrane. Without these structures, theorizing about how Bt toxins work is fantasy. Recent advances in electron cryo-microscopy (cryo-EM) have made these structures attainable. It is now time to attain them.

## Figures and Tables

**Figure 1 insects-12-00389-f001:**
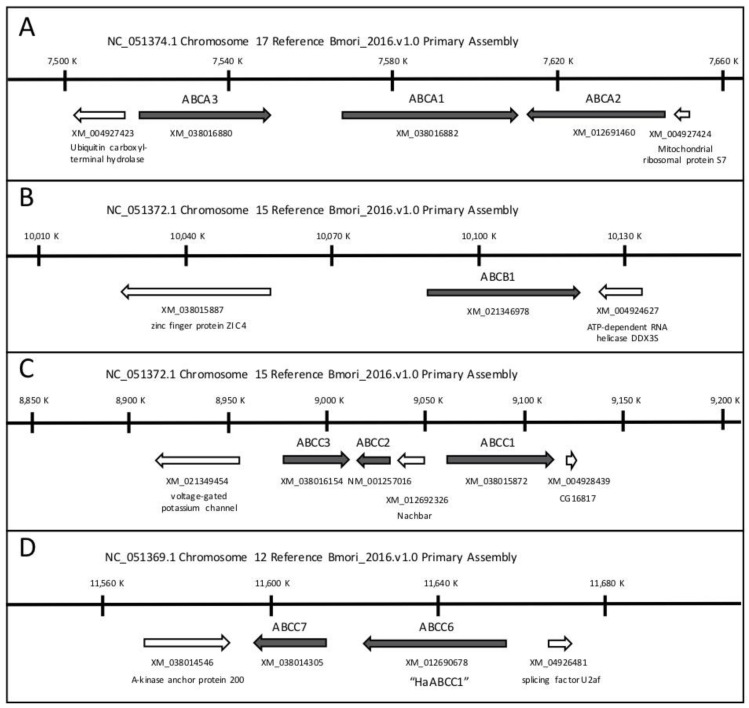
Genomic location of *Bombyx mori* (**A**–**C**) transporters orthologous to those interacting with three-domain Cry proteins described in the text. Coordinates are based on the chromosome view on NCBI (https://www.ncbi.nlm.nih.gov/, accessed on 2 April 2021) and are the same as shown in the new version of Kaikobase [9]. (**D**) Names for ABCC6 and ABCC7 are defined here for the first time, to avoid confusion with the previously named ABCC1.

**Figure 2 insects-12-00389-f002:**
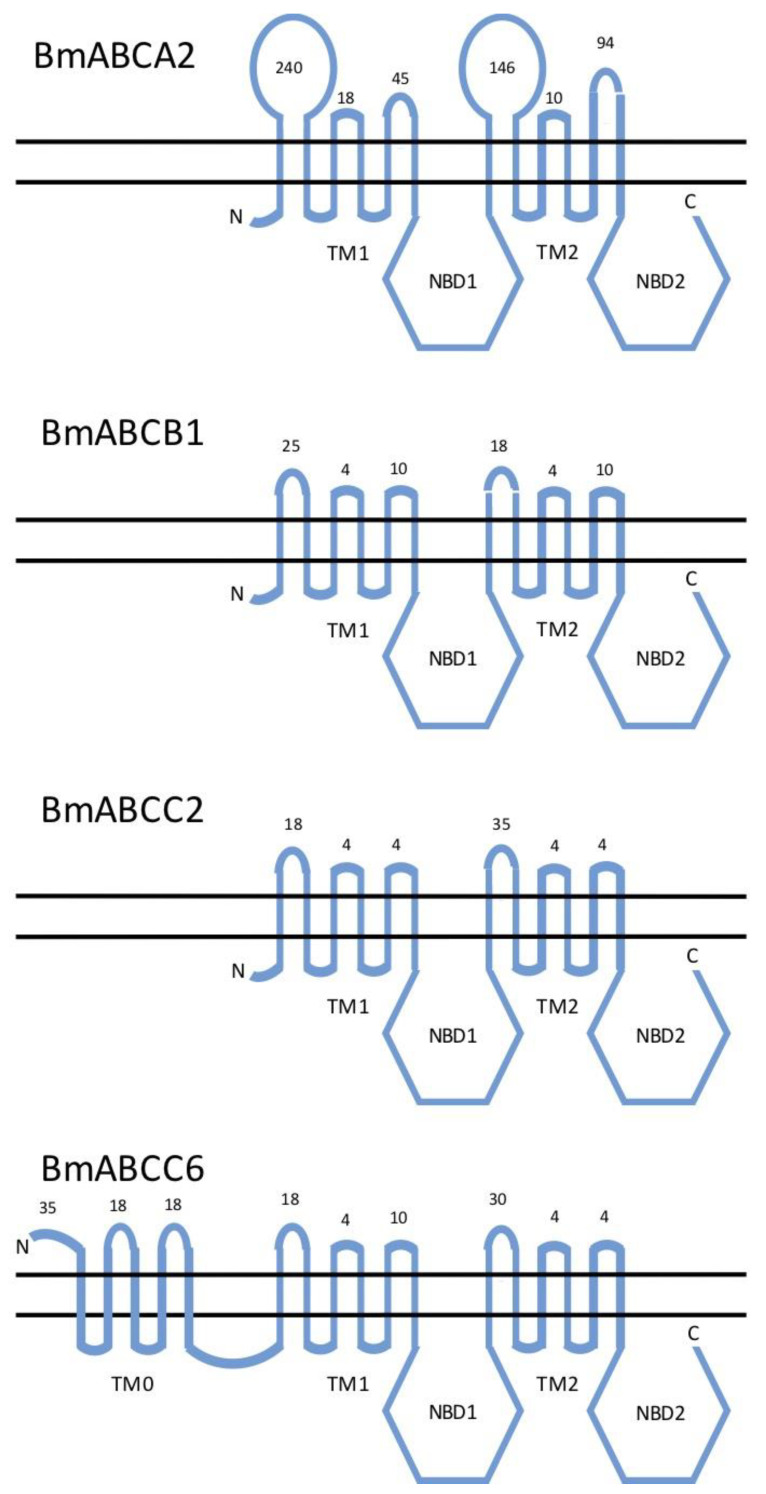
Phobius [10,11] predictions of membrane topology in the lipid bilayer of four ABC transporters from *Bombyx mori* orthologous to those interacting with three-domain Cry proteins described in the text (not to scale). The external lumenal surface is on top. Numbers of residues in each predicted external loop are shown. Transmembrane domains: TM0, TM1, and TM2. Nucleotide-binding domains: NBD1, NBD2. The N-terminus and C-terminus of each polypeptide is indicated.

**Figure 3 insects-12-00389-f003:**
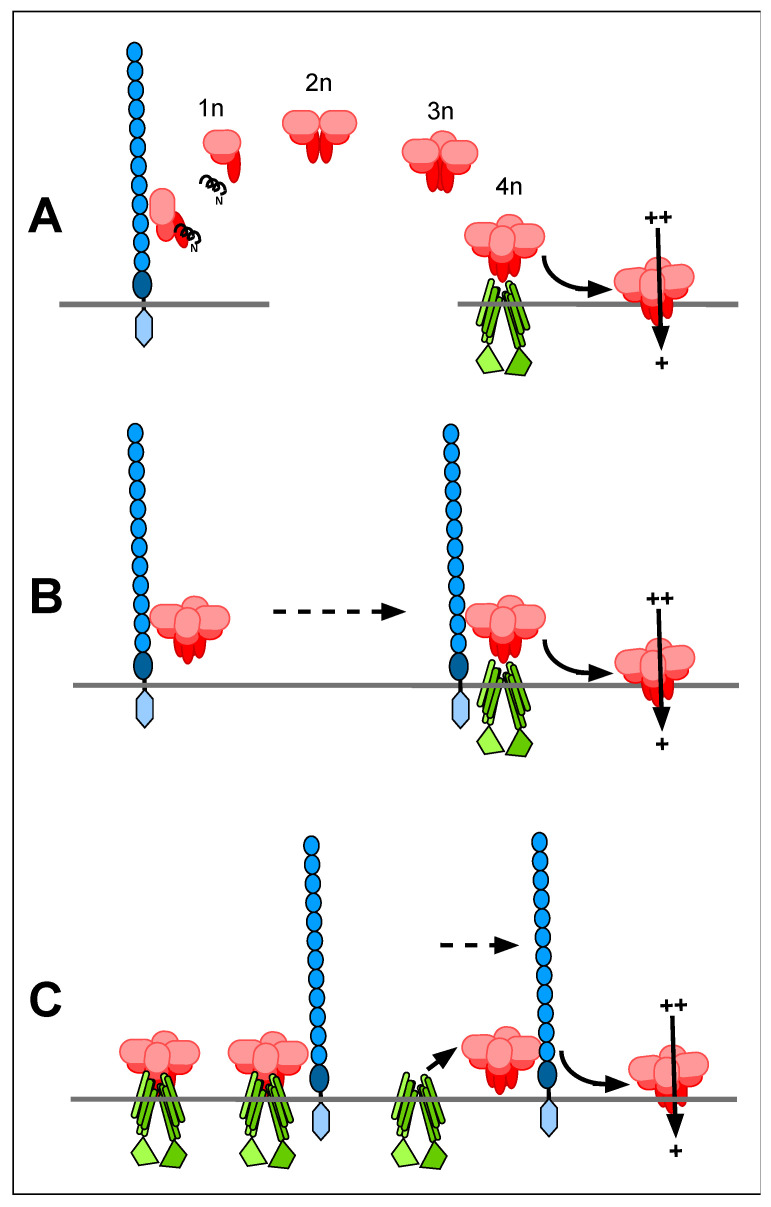
Hypothetical models of the mechanism of synergy between the 12-cadherin domain protein and an ABC transporter. (**A**) *Trans*-acting synergy, due to acceleration of oligomer formation following monomer binding to the cadherin. (**B**) *Cis*-acting synergy, due to the cadherin trapping the pre-pore and moving it to the ABC transporter. (**C**) *Cis*-acting synergy, due to the cadherin pulling the pre-pore away from the ABC transporter, freeing it to interact with another toxin pre-pore. Dotted lines represent movement of the cadherin within the membrane.

## Data Availability

NCBI GenBank: www.ncbi.nlm.nih.gov, accessed on 2 April 2021. Kaikobase: kaikobase.dna.affrc.go.jp, accessed on 2 April 2021.

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
