# Peer review of "The Essential and Enigmatic Role of ABC Transporters in Bt Resistance of Noctuids and Other Insect Pests of Agriculture"

_insects, 2021, doi:10.3390/insects12050389_

Round 1

Reviewer 1 Report

The manuscript ‘The role of ABC transporters in Bt resistance of noctuids and other insect pests of agriculture’ reviewed the most up-to-date of our understanding ABC transporters in insects and their contribution to insecticide resistance. The ATP-binding cassette (ABC) transporters are a family of large proteins in membranes and can transport a variety of compounds through membranes at the cost of ATP hydrolysis.  ABC genes are essential for many processes in the cell and represented in all extant phyla, from prokaryotes to eukaryotes. In humans. Mutations of ABC genes cause or contribute to several human genetic diseases. In insects, several resistances to Cry toxins have been contributed by the ABC transporter genes.  In this review, the author started the introduction of the ABC transporter gens, their membrane topology and their interaction with three-domain Cry proteins. The short introduction provided precise background knowledge for the general audience of the topic. Further, the review gives much detail about the evidence of Cry toxins resistance caused by mutation of ABC transporter genes followed the timeline of the initial discoveries of the Bt toxin resistance in insect. the search for the ABC mutants in resistant strains, the functional studies of heterologous expression, the interactions with the three-domain Cry toxins in the extracellular loops, the gene expression regulation and expression in cell lines, finally to the CRISPR/Cas9 Knockouts studies in insect species. In addition, the author provides the hypotheses on the mechanism of pore insertion and gives an insight into the future research of the ABC reporter genes in insects and its importance of resistance mechanisms. I would recommend highly for the manuscript to be published in Insect. I only have one suggestion for the author to add the recent work by Guan et al (2020) in Spodoptera frugiperda, they found a 12-bp insertion mutation in exon 15 of the ABCC2 gene from the Brazilian population.

Guan F, Zhang J, Shen H, Wang X, Padovan A, Walsh TK, Tay WT, Gordon KHJ, James W, Czepak C, Otim MH, Kachigamba D, Wu Y. (2020) Whole-genome sequencing to detect mutations associated with resistance to insecticides and Bt proteins in Spodoptera frugiperda. Insect Sci. Jun 18. doi: 10.1111/1744-7917.12838. Epub ahead of print. PMID: 32558234.

Author Response

Thank you for pointing out the reference by Guan et al. 2020 which I missed.  I have added it to the reference list, and cited it in Section 3.

Reviewer 2 Report

This is a well written review manuscript on the role of ABC transporters in BT resistance in important groups of  insect pests. The author did a great job is summarizing recent research outputs providing evidence of ABC transporters as a target of Bt toxins. 

The manuscript will benefit immensely from inclusion of a table that highlights the different categories of Bt toxins, their target pest species and document field-evolved resistance and also the geographic spread.

A schematic diagram illustrating the synergism f Cadherin and ABCC2 should also be included.

Author Response

Response:  With a five-day deadline for revising the manuscript, unfortunately I do not have time to construct a table with all the information that the reviewer has suggested. Morever, that would only duplicate information reviewed more comprehensively elsewhere.  I have already introduced the main classes of toxins in the introduction, which I believe is sufficient background to understand the interactions with ABC transporters.  For the reader who desires more information, I have added specific citations to Ref. 8, Adang et al. 2014 for a comprehensive list of toxins, and to Ref 6, Tabashnik et al. 2015 for a comprehensive treatment of the geographic spread of field-evolved resistance.

I have added a new Figure 3 to diagram hypotheses about the mechanism of synergism between cadherin and ABCC2. 

Reviewer 3 Report

The review by Heckel on ‘The role of ABC transporters in Bt resistance of noctuids and other insect pests of agriculture’ is a well written peace, argued from almost all perspectives, covering major topics ranging from the discoveries to mechanisms and mode of actions, tools and technologies needed to further understand the role of ABC transporters in Bt resistance in agriculturally important pests. While the review has very much harvested the low-hanging fruits that is to know about ABC transporters and Bt resistance (except the loopholes listed in the last lines of the ‘Future Perspectives), one can only wonder, is the Bt resistance among vital agricultural pests only the tip of the ice-berg for other microbial biopesticides!!

I have a minor concern, the review is well written and structured and it was very interesting and educational to read through. I feel as though; the ‘title’ of the review is not doing any justice to the content presented in the main text. The title covers the main body, but a couple more words to point out the essential and important ideas portrayed in the ‘future perspectives’ in my humble opinion would attract more readership.

Author Response

I have added three words to the title which I hope achieves the goal of attracting more readership.  The title now reads

"The essential and enigmatic role of ABC Transporters in Bt resistance of noctuids and other insect pests of agriculture"